# RoleAgent: Building, Interacting, and Benchmarking High-quality Role-Playing Agents from Scripts

**Jiaheng Liu\*[1], Zehao Ni\*[3,6], Haoran Que\*[2], Tao Sun[2], Zekun Wang[2], Jian Yang[2],**
**Jiakai Wang[3], Hongcheng Guo[2], Zhongyuan Peng[3], Ge Zhang[4], Jiayi Tian[2],**
**Xingyuan Bu[5], Ke Xu[2], Wenge Rong[2], Junran Peng[†,3,6], Zhaoxiang Zhang[1,6]**
[1]Nanjing University, [2]Beihang University, [3]University of the Chinese Academy of Sciences,
[4]University of Waterloo, [5]Beijing Institute of Technology,
[6]Institute of Automation, Chinese Academy of Sciences

## Abstract

Believable agents can empower interactive applications ranging from immersive environments to rehearsal spaces for interpersonal communication. Recently, generative agents have been proposed to simulate believable human behavior by using Large Language Models. However, the existing method heavily relies on human-annotated agent profiles (e.g., name, age, personality, relationships with others, and so on) for the initialization of each agent, which cannot be scaled up easily. In this paper, we propose a scalable RoleAgent framework to generate high-quality role-playing agents from raw scripts, which includes building and interacting stages. Specifically, in the building stage, we use a hierarchical memory system to extract and summarize the structure and high-level information of each agent for the raw script. In the interacting stage, we propose a novel innovative mechanism with four steps to achieve a high-quality interaction between agents. Finally, we introduce a systematic and comprehensive evaluation benchmark called RoleAgentBench to evaluate the effectiveness of our RoleAgent, which includes 100 and 28 roles for 20 English and 5 Chinese scripts, respectively. Extensive experimental results on RoleAgentBench demonstrate the effectiveness of RoleAgent.

## 1 Introduction

In cognitive models [6] and virtual environments [21, 3], researchers and practitioners have envisioned computational agents that can serve as believable proxies of human behavior. Such simulations of human behavior could populate virtual spaces and communities with realistic social phenomena [11, 31], test social science theories [4, 19], and underpin open world non-playable characters [21, 32].

Recently, Large Language Models [5] (LLMs) are used to simulate human behaviors at a single time point [31, 17], and the Generative Agents [30] in Fig. 1(a) produce agents that can *retrieve* relevant events and interactions over a long period, *reflect* on those memories to draw higher-level inferences, and reason to create *plans and reactions* that make sense at the moment and the longer-term arc of the agent's behaviors. However, these generative agents heavily rely on human-annotated agent profiles (e.g., name, age, personality, relationships with others, and so on) for the initialization of each agent, which influences the scalability to scale up the number of agents a lot. In contrast, in Fig. 1(b), we propose a more flexible agent framework called **RoleAgent** to automatically produce high-quality agents from the existing unprocessed scripts without using any human efforts on the initialization of

---

\* First three authors contributed equally.
† Corresponding Author: Junran Peng.

38th Conference on Neural Information Processing Systems (NeurIPS 2024) Track on Datasets and Benchmarks.

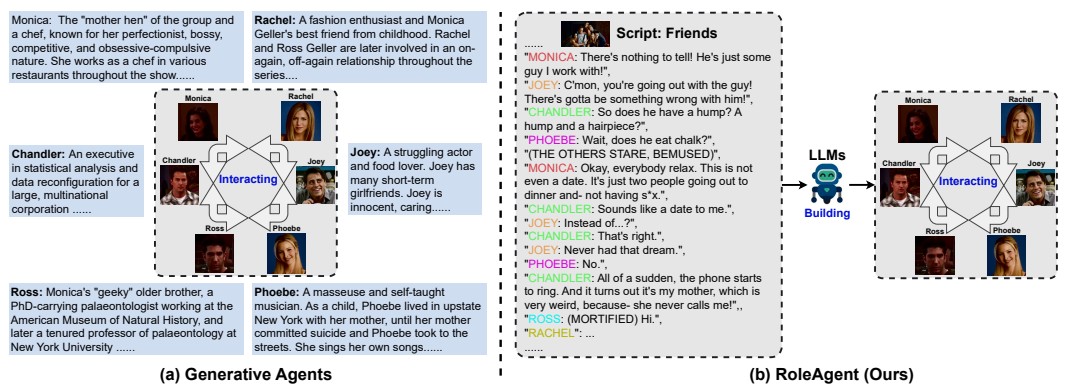

Figure 1: We take the script "Friends" as an example to show the differences between Generative Agents [30] in (a) and our RoleAgent in (b).

agents. The RoleAgent mainly includes two components: *how to build the RoleAgent* and *how to interact with RoleAgent*. Note that there are a large number of scripts from all kinds of agents, which indicates that the number of agents can **be scaled up easily** for RoleAgent.

Specifically, during the building stage, RoleAgent first undergoes a detailed extraction and summarization of structural and high-level information from raw unstructured scripts, which produces the initial observation of each agent. Then, as the initial observations of agents are typically redundant with sparse useful information, we further introduce the hierarchical memory scheme to distill and deduct existing raw observations into more structured memories. In the interacting stage, four steps (including query deconstruction, memory retrieval, memory summarization and response generation) are proposed to generate high-quality interaction experiences. Specifically, we introduce an innovative mechanism to update memory, facilitating nuanced interactions between agents. This mechanism incorporates processes for memory retrieval, caching, and replay. Meanwhile, we propose the dynamic importance score based on the retrieved frequency to represent the relevance between the memory and the interaction queries.

To rigorously assess the performance of RoleAgent, we have developed a specialized and extensive evaluation benchmark, named **RoleAgentBench**. Specifically, based on our constructed RoleAgentBench, we conduct two evaluations by "interviewing" the RoleAgent in natural language to probe the agents' ability to stay in character, remember, plan, react, and reflect accurately, which include agent evaluation and memory evaluation. For the agent evaluation, we perform a controlled evaluation to test whether the agents produce believable individual behaviors, where the self-knowledge, reaction and general abilities are evaluated. For the memory evaluation, we analyze the summarization quality.

Overall, the contributions are as follows: (1). We propose a flexible RoleAgent framework to automatically produce creative and interactive agents from raw scripts, which includes building and interacting stages and reduces the efforts of human-annotated agent profiles. (2). In the building stage, we propose the hierarchical memory system to reason and store structural and high-level memories of different roles. In the interacting stage, we introduce an innovation mechanism with four steps to obtain sufficient context and generate high-quality responses. (3). To evaluate RoleAgent, we introduce a comprehensive evaluation benchmark called RoleAgentBench including 128 roles from 20 English and 5 Chinese scripts, and extensive experiments show the advantages of RoleAgent.

## 2 Related Works

**LLM-based Agents** The evolution of Large Language Models (LLMs) [36, 37, 27, 46, 2, 16, 48, 25, 24] as core controllers in autonomous agents have led to significant advancements in their ability to carry out complex tasks. Early attempts like Auto-GPT [44] demonstrated the potential of using LLMs for goal-oriented tasks without multi-agent collaboration. To address this limitation, systems like BabyAGI [35] and MetaGPT [18] were introduced, showcasing how assigning specific roles to a group of LLMs can facilitate coordination toward common objectives, such as software development.

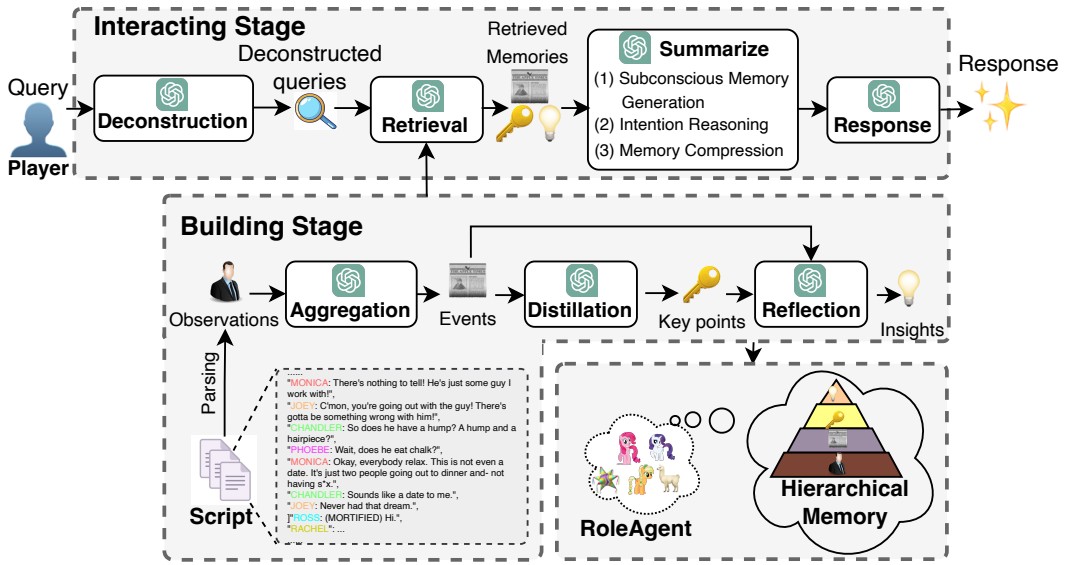

Figure 2: The overall framework of RoleAgent can be divided into two workflows: (1) building RoleAgent from scripts; (2) interacting with RoleAgent (as a player or a non-player character (NPC). Note that for simplicity, we will use the term "player" as a general designation for interaction.
.

Meanwhile, some researchers [22, 8, 23, 39, 42] illustrated the benefits of communicative role-playing in completing tasks and tried to utilize external tools. Despite these advancements, a common constraint in these systems is the reliance on predefined or manually created agents with set roles, which can potentially restrict the breadth of collaborative applications.

**Believable Agents**   Believable agents [20, 33, 38, 41, 40, 34] are essential in crafting immersive interactive experiences, serving to emulate the nuanced decision-making and charm of characters typical in animations or sophisticated game NPCs. These agents [9, 45, 28] are intended to populate virtual spaces, replicating human-like behaviors to enable narratives and social dynamics to unfold naturally. Besides, learning-based techniques, especially reinforcement learning, have been effective in honing agent behavior within competitive gaming areas [14, 15]. Creating non-competitive agents that can move through open worlds and interact socially remains challenging [43]. Recently, [30] applies the LLMs to generate believable agents with a memory stream. In contrast, our RoleAgent aims to produce and evaluate agents automatically from massive raw scripts without human effort.

## 3   RoleAgent

### 3.1   Overall Framework

In Fig. 2, before deploying RoleAgent in an interactive game environment, we need to build the agents from scripts. This **building stage** involves the structuration and summarization of script content into higher-order, dense memories. Unlike Generative Agents [29] which needs extensive human efforts to define the agent profiles and periodically summarizes the agents' observations, RoleAgent involves a recursive process of extraction and abstraction, organizing and distilling low-level observations with a hierarchical memory, where lower-level memories are detailed and sparse, while higher-order memories are abstract and dense. Then in **interacting stage**, the queries of player[1] are first deconstructed to facilitate multi-faceted memory retrieval. The retrieved memories are subsequently summarized from various perspectives to formulate LLM contexts, enabling RoleAgent to perform actions or give responses with strong role-specific knowledge and episodic memories. In

---

[1]"Player" denotes human or other produced RoleAgents.

the following sections, we will delineate the workflow of RoleAgent. For the sake of clarity, we will employ the character of "Iron Man" as a prototypical RoleAgent.

## 3.2 Building Stage

In Fig. 2, the building stage involves two steps : (1) script parsing, the extraction of an agent's observations from scripts; (2) hierarchical memory construction, a recursive distillation and deduction of observations into a structured memory system.

**Script Parsing**   As Scripts are often unstructured text, it is imperative to parse the scripts into formats that are both structured and queryable. Given a script, we initially identify the characters' names and subsequently extract all behaviors and dialogues attributable to each character. This data is formatted as "`{<character>:  <behavior or dialogue>}\n{<character>: <behavior or dialogue>}...`". It is important to note that asides are categorized as a "special character" to incorporate essential background or plot information. These elements are chronologically arranged and segmented in alignment with the natural plot segmentation of the script.

When focusing on several particular roles, the role profiles are defined as the observations of these roles. To extract the observations, we first cut the raw script into scenes arranged in chronological order. Then we determine the relevant roles involved in each scene based on GPT-4. Finally, we select several roles from the script based on their participation and popularity, and generate role profiles based on dialog and narration.

**Hierarchical Memory**   During script parsing, the role's perceivable information is meticulously compiled. However, these raw observations are typically redundant, and the useful information is sparse, which hampers the agent's ability to rapidly and accurately retrieve relevant memories. Moreover, overly granular information impedes generalization. For instance, it would not be feasible to extrapolate the real relationship between Iron Man and Captain America solely from observations of their combat interactions. Inspired by the theories of information hierarchy in the human brain [10, 7, 49], we begin with observations and abstract them progressively through a three-tiered process: (1) event aggregation; (2) key point distillation; (3) insight reflection. The observations along with the resultant events, key points, and insights, constitute the **hierarchical memories**, where the events, key points, and insights are combined as **high-order memories**.

Specifically, from $K_o$ observations, we engage GPT-4 to aggregate these into an event. This is achieved by presenting instructions such as "`What happened in these observations?`". After several iterations, we obtain $K_e$ events. Note that $K_e$ is equal to the number of scenes. Subsequently, the LLM is prompted to distill these events into a key point by instructions like "`What do these events illustrate?`" Finally, we ask the LLM to reflect an overarching insight from $K_p$ key points and their corresponding events, using the instruction like "`What is the overarching insight that can be drawn from these key points and their relevant events?`". Note that we set $K_p = 3 \times K_e$, which means that we extract 3 key points for each event. Through this structured process, observations are progressively abstracted into events, key points, and ultimately, insights. For example, Iron Man may perceive a series of observations like "`{"Captain America":  "(Punched Iron Man in the face)}\n{"Iron Man":  "Why did you kill my father?"}\n...`". These observations can be aggregated into an event described as "`Iron Man and Captain America had a fight regarding to the death of Iron Man' father.`" Subsequently, another event may occur: "`Iron Man and Captain America had apologized for each other.`" These two events can then be distilled into a key point: "`The contradiction between Iron Man and Captain America about the death of his father is resolved.`"  Coupled with other key points such as "`Iron Man and Captain America fight together for the Earth.`"  and all the supporting events, we can derive the insight "`Iron Man prioritizes the survival of the planet.`"

## 3.3 Interacting Stage

In Fig. 2, the interacting stage with a pre-built RoleAgent can be divided into four steps: (1) query deconstruction (2) memory retrieval, (3) memory summarization, and (4) response generation.

**Query Deconstruction**   For a RoleAgent $A$, when a player $P$ presents a query $q$, the query is first deconstructed into three variants: (1) "`Who is` $A$?" ($q_1$), (2) "`Who is` $P$?" ($q_2$), and (3) "$P$ `said:` $q$" ($q_3$), which aims to analyze the positions of the player $P$ and the RoleAgent $A$ and the rationale of the query $q$ and provide additional cues for further interactions.

**Memory Retrieval**   Given deconstructed queries, we retrieve the relevant high-order memories from events, key points, and insights and produce corresponding memories: $M$. and then the initially retrieved memories are re-ranked based on the following scores. [2]

We assign four distinct scores to each item in the high-order memories: (1) a dynamic importance score $s_{di}$, (2) a static importance score $s_{si}$, (3) a timeliness score $s_t$, and (4) a correlation score $s_c$. Specifically, first, the dynamic importance score is the frequency with which memory is retrieved throughout the interaction stage, denoting the relevance between the memory and the actual interaction queries. Second, following [30], the static importance score is to enable the LLM to assess the generic importance of a particular memory, where the score is from 0 to 5. Third, timeliness score assigns a higher score to memories that were recently accessed, and we employ an exponential decay function with a decay factor of 0.99 following [30]. Fourth, the correlation score assigns a higher score to the most relevant memories by computing the cosine similarity of embedding vectors of memories. Finally, we normalize these four scores to the range of [0, 1] by min-max scaling, and calculate the overall score $s$ for each memory as follows: $s = s_{di} + s_{di} + s_t + s_c$. This mechanism ensures that the most relevant, important, and timely memories are utilized in response to the player's query.

To provide a more tangible memory context, we propose the **memory replay** to replay the observations $O$ indexed from the retrieved high-order memories, and combine these observations with the high-order memories $M$ to generate the refined memories $M'$, which are sent to LLMs.

**Memory Summarization**   The refined memories $M'$ are fed into LLMs for summarization and five distinct instructions are used: three are aimed at eliciting subconscious memories, one is to reason the player's intention, and the final instruction is to compress the relevant content. First, we build **Subconscious Memories**. Specifically, the subconscious memories encompass three categories of information: (1) the character traits and position of the RoleAgent $A$, (2) the character traits and position of the player $P$, and (3) the relationship between $A$ and $P$. To facilitate summarization, we employ specific instructions: for $A$, we use "`Summarize` $A$`'s character traits and position, avoiding embellishment.`", for $P$, the instruction is "`Summarize` $P$`'s character traits and position, avoiding embellishment.`", and to elucidate their relationship, we prompt with "`Infer and briefly describe the relationship between` $A$ `and` $P$`.`". These instructions are applied to $M'$ to generate summaries, which are then concatenated to form the subconscious memories, denoted as $C_{sub}$. Second, we apply **Intention Reasoning and Memory Compression**. Specifically, we prompt LLMs to reason about $P$'s intention with the instruction "`Why does` $A$ `think` $P$ `said:` $q''$?", using the refined memory set $M'$. The generated response is denoted as $C_{int}$. Moreover, we use the instruction "`Summarize the relevant content of` $P$ `said:` $q''$.`" for LLMs to compress $M'$ into $C_{rel}$.

**Response Generation**   We concatenate $C_{sub}$, $C_{int}$, $C_{rel}$, and "$A$ `observed` $P$ `said:` $q''$.`" with the task instruction "`Please generate a response`", using a specific prompt template. This assembled prompt is to instruct the LLM to generate a response with role-specific knowledge and episodic memories of the RoleAgent for the player.

# 4   RoleAgentBench

To evaluate RoleAgent, we construct the RoleAgentBench including 128 roles from 5 Chinese and 20 English scripts. Besides, our RoleAgentBench evaluates two aspects (i.e., the qualities of the overall agent simulation and the specific memory system) with 4 subtasks, and we illustrate the construction details as follows. Note that all questions and answers are generated based on the script and GPT-4, which are then revised by human annotators. See Appendix C for samples of RoleAgentBench.

---

[2]Note that the Faiss library [12] is used for efficient similarity search in retrieval.

### 4.1 Agent Simulation

To evaluate whether RoleAgent simulates roles well, we evaluate the following three parts.

**Self-Knowledge**: Self-Knowledge tests the Agent's ability to recognize its attributes in the form of true or false questions format, in which the Agent has to judge the four questions related to itself. These questions focus on the occupation, relationships, name, and personality, where each question has a corresponding reference answer (True or False). We use the **accuracy** for Self-Knowledge.

**Reaction**: Reaction tests the Agent's ability to react to responses for different roles. For example, given the same question, a specific Role A will generate different answers for different roles based on the relationships or positions between Role A and other roles. We use the **accuracy** for reaction.

**General Response**: General Response tests the Agent's general communication ability in question-answer format. Role A asks a question to role B, and RoleAgent needs to simulate role B to reply to the question. Each question has a reference answer, which is highly accurate and stylized for role B. **Win rates** are reported based on human and GPT-4, where 3 human annotators are employed.

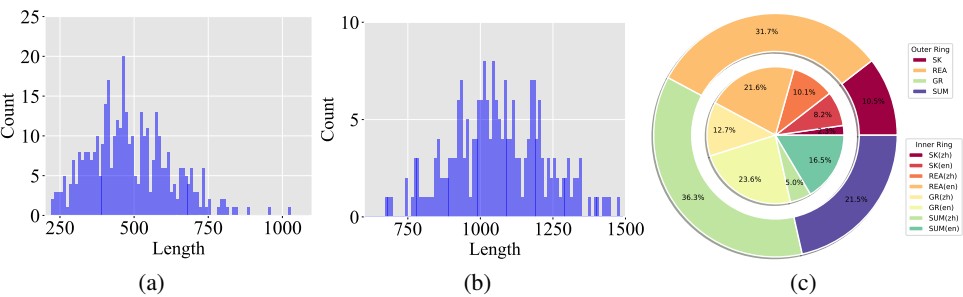

Figure 3: (a). General Response. (b). Summarization. (c). Distribution of subtasks.

### 4.2 Memory System

To test the capabilities of the memory system, we mainly evaluate the summarization qualities.

**Summarization**: As summarization is a high-density content, we evaluate the **entity density (ED)** of the generated summary by extracting the entities of the summary and dividing the number of entities by the summary length. Higher entity density denotes a higher information density. We also obtain the **entity recall, (ER)** between the entities of the generated summaries and the golden summary entities, where higher recall indicates higher qualities. Besides, we report the ER/ED results to denote the ratio of valid entities. Meanwhile, **win rates** using GPT-4 and human are also reported.

### 4.3 Statistic Analysis

In Table 6 of Appendix C, we provide the number of samples on 20 English and 5 Chinese scripts of each task. In Fig. 3(a) and Fig. 3(b), we provide the length distribution of the addition between questions and answers for general response and summarization tasks. In Fig. 3(c), the ratios of all questions for each script are provided. Note that "SK", "REA", "GR", "SUM" denote "self-knowledge", "reaction", "general response" and "summarization", respectively.

## 5 Experiments

### 5.1 Main Results

In Table 1 and Table 2, we provide the results of our RoleAgent based on different LLMs, where GPT-4 [27], Qwen-Max [2], GPT-3.5 [26], Yi-34B [47], Qwen1.5-14B [2], Mistral-7B [3], LLaMA3-8B [1], ChatGLM3-6B [13] and RoleLLM [41] are used as baselines. Note that we follow RoleLLM to reproduce RoleLLaMA and RoleGLM using the RoleBench dataset, and we use RoleLLaMA and

---

[3] https://huggingface.co/mistralai/Mistral-7B-Instruct-v0.2

Table 1: Performance on the English subset of RoleAgentBench. "WR-G", "WR-H", "ED", and "ER" are win-rate of GPT-4, win-rate of human evaluation, entity density, and entity recall, respectively.

| Models | Agent Simulation | | | | Memory System | | | | |
|---|---|---|---|---|---|---|---|---|---|
| | Self-Knowledge | Reaction | General Response | | Summarization | | | | |
| | Acc. | Acc. | WR-G | WR-H | ED | ER | ER/ED | WR-G | WR-H |
| **GPT-4** | **94.7** | 88.9 | **48.6** | 49.7 | 8.9 | 39.8 | **4.47** | **75.8** | 59.8 |
| **Qwen-Max** | 94.1 | **92.9** | 46.8 | **54.3** | 9.8 | 41.8 | 4.27 | 72.4 | **62.9** |
| **GPT-3.5** | 87.6 | 76.7 | 37.6 | 41.5 | 11.6 | 36.8 | 3.17 | 56.7 | 51.6 |
| **Yi-34B** | 88.5 | 73.8 | 43.5 | 42.5 | 11.3 | 41.1 | 3.64 | 55.1 | 44.7 |
| **Qwen1.5-14B** | 83.7 | 71.0 | 40.5 | 45.3 | 11.4 | 40.2 | 3.53 | 41.7 | 38.0 |
| **Mistral-7B** | 87.7 | 64.9 | 40.1 | 37.4 | 10.7 | 40.5 | 3.79 | 44.9 | 48.9 |
| **LLaMA3-8B** | 84.0 | 68.1 | 36.6 | 38.9 | 10.9 | 32.8 | 3.01 | 47.0 | 50.2 |
| **RoleLLM** | 61.0 | 31.0 | 25.3 | 15.2 | 17.2 | 23.7 | 1.38 | 25.4 | 10.7 |

Table 2: Performance on the Chinese subset of RoleAgentBench.

| Models | Agent Simulation | | | | Memory System | | | | |
|---|---|---|---|---|---|---|---|---|---|
| | Self-Knowledge | Reaction | General Response | | Summarization | | | | |
| | Acc. | Acc. | WR-G | WR-H | ED | ER | ER/ED | WR-G | WR-H |
| **GPT-4** | **92.5** | 73.9 | **44.8** | 39.7 | 10.0 | 45.0 | 4.50 | 75.1 | 69.5 |
| **Qwen-Max** | 90.3 | **84.7** | 39.4 | **41.5** | 8.8 | 44.7 | **5.08** | 79.6 | 71.4 |
| **GPT-3.5** | 84.8 | 61.3 | 34.9 | 35.6 | 14.9 | 38.3 | 2.57 | 53.7 | 61.5 |
| **Yi-34B** | 80.0 | 76.7 | 38.2 | 29.1 | 8.1 | 40.6 | 5.01 | 61.6 | 58.4 |
| **Qwen1.5-14B** | 82.7 | 73.8 | 38.7 | 33.7 | 11.1 | 43.1 | 3.88 | 67.0 | 62.3 |
| **ChatGLM3-6B** | 81.3 | 59.6 | 36.9 | 39.1 | 16.0 | 36.6 | 2.29 | 44.2 | 35.8 |
| **LLaMA3-8B** | 70.1 | 62.4 | 25.1 | 29.9 | 10.9 | 35.1 | 3.22 | 43.4 | 42.0 |
| **RoleLLM** | 72.1 | 59.3 | 20.6 | 25.7 | 16.0 | 35.3 | 2.21 | 39.2 | 45.1 |

RoleGLM to test the English and Chinese subsets of RoleAgentBench, respectively. Our analysis led to the following key findings: (1) We observe that API-based models (e.g., GPT-4 and Qwen-Max) achieve significant improvements when compared to these open-source models (e.g., Yi-34B and Qwen1.5-14B), which shows that the capacities of base models influence the agent's abilities a lot. (2) The scores on the Chinese subset of English-centric LLM (i.e., LLaMA3-8B) are lower than the corresponding results on the English subset. In contrast, the bilingual (English-Chinese) LLMs (e.g., Qwen1.5-14B, Yi-34B) achieve comparable results on English and Chinese subsets. (3) We observe that RoleLLM fine-tuned on the role-related dataset cannot perform well in our dataset. We suppose the RoleLLM only focuses on improving the speaker style of the simulated roles, which is essentially different from the four subtasks in RoleAgentBench. (4) These well-performed LLMs (e.g., GPT-4 and Qwen-Max) obtain lower ED and higher ER/ED results, which means that the number of entities is relatively smaller but the number of valid entities is higher than the results of other LLMs, which further shows the effectiveness of powerful LLMs.

## 5.2  Ablation Study

To show the effect of our hierarchical memory system and memory replay for memory retrieval, we perform experiments in Table 3 based on GPT-3.5. Note that "HM" and "MR" denote hierarchical memory and memory replay, respectively. In Table 3, when removing the memory replay, the performance results degrade, which shows that it is beneficial to use the raw extracted observations related to the events or insights for better results. Moreover, when removing the hierarchical memory system, the performance results degrade a lot, specifically the self-knowledge and reaction sub-tasks, which demonstrates the advantage of the hierarchical memory system.

Table 3: Performance on the English subset of RoleAgentBench.

| Models | Agent Simulation | | | Memory System | | | |
|---|---|---|---|---|---|---|---|
| | Self-Knowledge | Reaction | General Response | Summarization | | | |
| | Acc. | Acc. | WR-G | ED | ER | ER/ED | WR-G |
| **RoleAgent** | **87.6** | **76.7** | **37.6** | 11.6 | 36.8 | 3.17 | **46.7** |
| w/o MR | 73.6 | 76.4 | 35.2 | 10.4 | 34.7 | **3.34** | 35.4 |
| w/o MR&HM | 70.3 | 54.5 | 32.2 | 11.3 | 35.7 | 3.16 | 29.5 |

Table 4: Performance on English subset of RoleAgentBench. "**Gen. Agents**" is "Generative Agents".

| Models | Agent Simulation | | | Memory System | | | |
|---|---|---|---|---|---|---|---|
| | Self-Knowledge | Reaction | General Response | Summarization | | | |
| | Acc. | Acc. | WR-G | ED | ER | ER/ED | WR-G |
| Gen. Agents | 68.9 | 40.7 | 31.8 | 5.1 | 20.7 | 4.06 | 38.9 |
| **RoleAgent** | 87.6 | 76.7 | 37.6 | 11.6 | 36.8 | 3.17 | 46.7 |

## 5.3 Further Analysis

**Compare with Generative Agents.** We use the GPT-3.5 as the baseline LLM to compare our RoleAgents with Generative Agents (**Gen. Agents**) [30] on the English subset of RoleAgentBench in Table 4. Note that Generative Agents heavily rely on extensive human efforts to produce the role profiles. In Table 4, we observe our RoleAgent is better than Generative Agents a lot on these subtasks. For example, our RoleAgent can predict self-knowledge well without using any human efforts, which means that RoleAgent can understand the basic attributes (e.g., career or relationships) of simulated roles well. Second, our RoleAgent produces better reaction and summarization abilities, where can be attributed to the intention reasoning strategy and subconscious memories in Sec. 3.3.

**Analysis on common and uncommon scripts.** We take GPT-3.5 as the baseline LLM to analyze the results on several common and uncommon scripts in Table 5, which aims to clarify if the LLM is inferring from scripts or recalling information stored in model weights. Specifically, for common scripts, we select "Harry Potter" and "Friends". for uncommon scripts, we select "Alias" and "Degrassi Next Generation". Then, for the GPT-3.5 (Internal), we just prompt the GPT-3.5 to generate the answer to all questions of these subtasks without using any additional information. In contrast, GPT-3.5 (External) is our RoleAgent based on GPT-3.5, which can infer from scripts to obtain additional contexts. In Table 5, first, we observe that the results of GPT-3.5 (External) using RoleAgent are better than GPT-3.5 (Internal) a lot on both common and uncommon scripts, which means that RoleAgent can simulate agents well based on our hierarchical memory. Second, we observe that results of GPT-3.5 (Internal) drop a lot on uncommon scripts on self-knowledge and reaction, which shows it is necessary to use RoleAgent for simulating different roles.

**Visualization on the hierarchical memory of building stage.** We take the script of Harry Potter as an example to show the process of producing the hierarchical memory of RoleAgent. In Fig. 4(a), we provide extracted high-order hierarchical memories with high scores after the initialization procedure on the Harry Potter script. See Fig. 7 in Appendix D for more details on the building stage.

**Visualization of each step of the interaction stage.** In Fig. 5, we visualize four interaction steps by playing with RoleAgent "Harry". Note that the human plays the role of "Hermione". Given the query of "Hermione", our RoleAgent "Harry" produces a high-quality and interesting response, which shows the effect of RoleAgent. See Fig. 12, Fig. 13 and Fig. 14 for more results in Appendix E.

**Visualization on the interaction between Human and RoleAgent.** We add the visualization by playing with RoleAgent "Sheldon". Note that the human plays the role of "Lenoard". Specifically, after the building stage of RoleAgents from "Scene: A corridor at a sperm bank" (See Fig. 15 of Appendix for more details on this scene of the script "Friends"), we begin to interact with the RoleAgent "Sheldon". The overall interaction process is shown in Fig. 16 of the Appendix E, and we take some interesting samples as shown in Fig. 4(b), where we highlight in the red box to show some high-quality examples of agent simulation for "Sheldon".

Table 5: Performance on the two common and two uncommon scripts of RoleAgentBench.

| Models | Agent Simulation | | | Memory System | | | |
|---|---|---|---|---|---|---|---|
| | Self-Knowledge | Reaction | General Response | Summarization | | | |
| | Acc. | Acc. | WR-G | ED | ER | ER/ED | WR-G |
| **Common Scripts** | | | | | | | |
| **GPT-3.5 (Internal)** | 82.5 | 57.6 | 29.4 | 10.7 | 16.4 | 1.53 | 20.5 |
| **GPT-3.5 (External)** | 90.4 | 77.8 | 45.1 | 12.2 | 37.8 | 3.10 | 47.2 |
| **Uncommon Scripts** | | | | | | | |
| **GPT-3.5 (Internal)** | 55.0 | 24.3 | 26.7 | 10.2 | 18.4 | 1.80 | 31.9 |
| **GPT-3.5 (External)** | 77.5 | 54.9 | 41.8 | 9.1 | 39.1 | 4.30 | 45.8 |

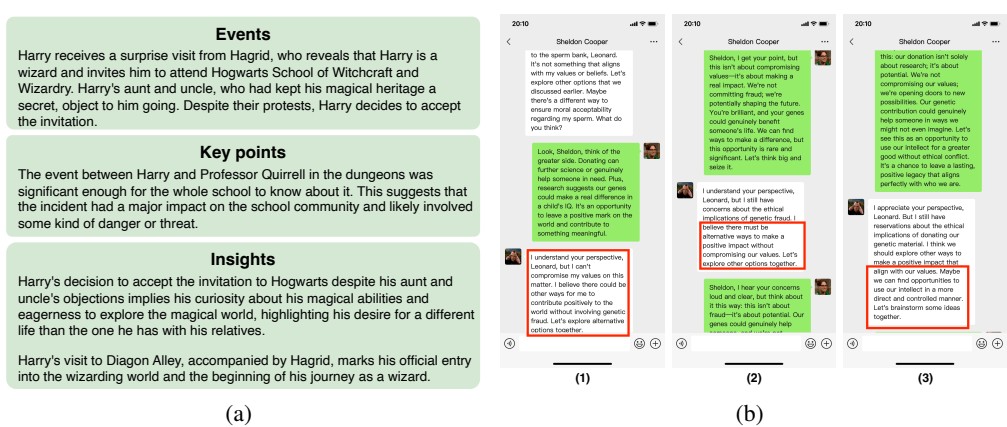

Figure 4: (a). Samples of events, key points and insights. (b). Interaction between human ("Lenoard") and RoleAgent ("Sheldon"). "(1)", "(2)" and "(3)" are "(2)", "(3)" and "(6)" of Fig. 16, respectively.

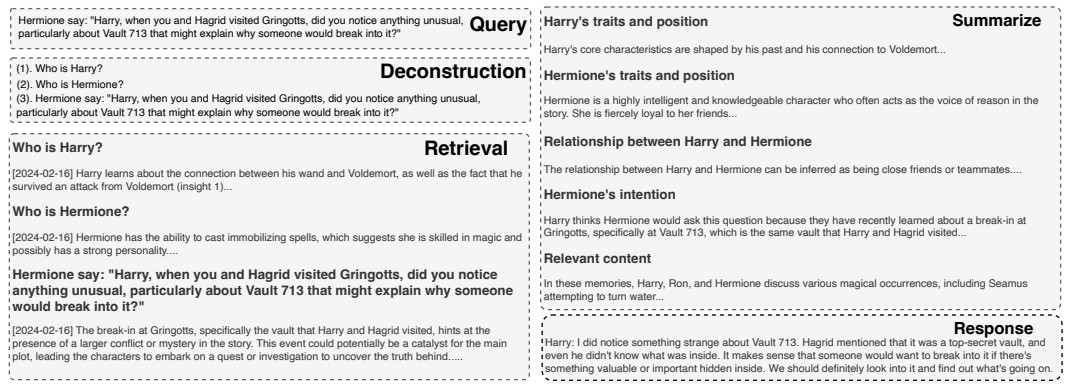

Figure 5: Interaction between human ("Hermione") and our RoleAgent ("Harry").

# 6 Conclusion

In this study, we introduce a framework named RoleAgent designed to construct agents directly from unprocessed scripts, which consists of building and interacting stages. This approach minimizes the need for manually created agent profiles and enables the autonomous generation of inventive and engaging agents. During the construction phase, we implement a hierarchical memory system to logically organize and preserve the structural and advanced memories associated with various characters. For the interaction phase, we deploy an innovative four-step mechanism to capture ample context and produce responses of superior quality. To assess the performance of RoleAgent, we have established an extensive evaluation benchmark termed RoleAgentBench. Extensive experiments on RoleAgentBench demonstrate the superior capabilities of the RoleAgent framework.

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

# A  Social Impacts and Limitations

Our RoleAgent could produce content that is sensitive or damaging. This is because it may mirror the aggressive, obscene, or prejudiced characteristics of specific personas. The development of this work, including all related materials, is intended for academic investigation. For the limitations, extensive costs (e.g., API costs and GPU consumption costs) are needed for building and interacting with RoleAgent.

# B  Details behind Building RoleAgentBench

## B.1  Self-Knowledge

Self-knowledge aims to evaluate a RoleAgent's ability to recognize its own attributes. To assess this, we manually design four true or false questions along with their reference answers (True or False) for each RoleAgent. These questions focus on the agent's occupation, relationships, name, and personality. During the evaluation process, the RoleAgent determines the correctness of these true or false questions. The output is required to be in JSON format to facilitate subsequent processing. We use Accuracy to represent the level of Self-Knowledge mastery.

## B.2  Reaction

Reaction aims to evaluate the ability of a RoleAgent to react to different roles. For different inquirers, the responses of a RoleAgent need to be based on factors such as their relationships or intentions. For example, even with the same question, Sherlock's responses to Watson and Moriarty would be vastly different. Refer to Figure 6 for a schematic illustration of the building Reaction subtask. First, I will generate responses from RoleAgent A to different inquirers based on GPT-4. Specifically, these questions are from the General Response subtask. In the evaluation phase, I will convert these responses and questions into multiple-choice questions, and then have RoleAgent A select an option to respond to each inquirer. The evaluation metric will also be Accuracy.

## B.3  General Response

General Response tests the general communication ability of a RoleAgent in question-answer format. In the Script Parsing phase, we have divided a complete script into multiple scenes. Based on GPT-4, we then generated questions and reference answers for each scene. After generation, we deduplicated these questions using ROUGE-L, removing the top 5% of data pairs with the highest ROUGE-L scores. Subsequently, we manually filter the data and fine-tune the responses. We use GPT-4 and human evaluators to evaluate the responses of different models and frameworks.

## B.4  Summarization

Summarization evaluates the agent's ability to summarize the retrieved content. We prompt GPT-4 with questions from General Response and whole script content to generate reference summaries

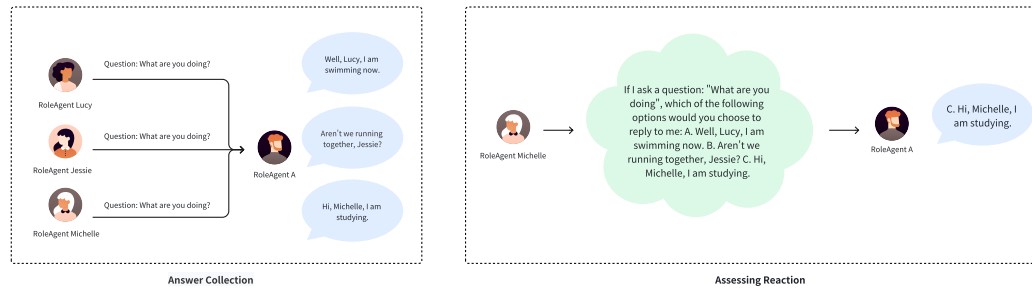

Figure 6: Illustration of building Reaction subtask.

of the questions. We then use ED, ER, ER/ED, and win rates to test the summarization abilities of different models and frameworks. We also prompt GPT-4 to extract entities from summaries.

# C   Details on RoleAgentBench

**Benchmark Link**:

https://huggingface.co/datasets/RoleAgent/RoleAgentBench

Twenty English scripts and the supported RoleAgents are as follows:

- **Merchant of Venice**: Antonio, Shylock, Bassanio, Portia
- Episode 1 of **Sherlock** Season 1: Jeff, John, Lestrade, Mycroft, Sherlock
- **Harry Potter** Season 1: Harry, Hermione, Malfoy, McGonagall, Ron
- Episode 1 of **The Big Bang Theory** Season 1: Howard, Leonard, Penny, Raj, Sheldon
- Episode 1 of **Friends** Season 1: Chandler, Joey, Monica, Paul, Phoebe, Rachel, Ross
- **Alias**: Dixon, Mr.Bristow, Syndey, Vaughn, Will
- **Bones**: Angela, Booth, Brennan, Hodgins, Zach
- Episode 1 of **Buffy the Vampire Slayer** Season 1: Buffy, Cordelia, Giles, Willow, Xander
- Episode 1 of **Charmed** Season 1: Andy, Jeremy, Phoebe, Piper, Prue
- **Degrassi Next Generation**: Ashley, Caitlin, Emma, Manny, Toby
- Episode 1 of **Frasier** Season 1: Daphne, Frasier, Martin, Niles, Roz
- **Game of Thrones**: Arya Stark, Catelyn Stark, Eddard Stark, Tyrion Lannister
- Episode 1 of **Glee** Season 1: Emma, Finn, Rachel, Terri, Will
- **Grey's Anatomy**: Cristina, Derek, George, Izzie, Meredith
- **Hannibal**: Abigail, Alana, Hannibal, Jack, Will
- Episode 1 of **How I Met Your Mother** Season 1: Barney, Lily, Marshall, Robin, Ted
- **Lucifer**: Charlotte, Chloe, Daniel, Lucifer, Maze
- **My Little Pony Friendship is Magic**: Applejack, Pinkie Pie, Rainbow Dash, Rarity, Twilight
- **Once Upon A Time**: Emma, Henry, Prince Charming, Regina, Snow White
- **Rick and Morty**: Beth, Jerry, Morty, Rick, Summer

Five Chinese scripts and the supported RoleAgents are as follows:

- 西游记(三打白骨精): 八戒, 白骨精, 黑狐精, 沙僧, 唐僧, 悟空
- 唐人街探案第一季: 阿香, 坤泰, 秦风, 思诺, 唐仁
- 九品芝麻官: 包龙星, 豹头, 常威, 方唐镜, 来福, 戚秦氏, 有为
- 狂飙(第一集): 安欣, 高启强, 李响, 唐小龙, 徐忠
- 家有儿女(第一集): 刘梅, 刘星, 夏东海, 小雪, 小雨

## C.1   Agent Simulation

**Self-knowledge** The samples of Self-knowledge are shown in Fig. 8.

**Reaction** The samples of Reaction are shown in Fig. 9.

**General Response** The samples of General Response are as shown in Fig. 10.

## C.2   Memory System

**Summarization** The samples of Summarization are shown in Fig. 11.

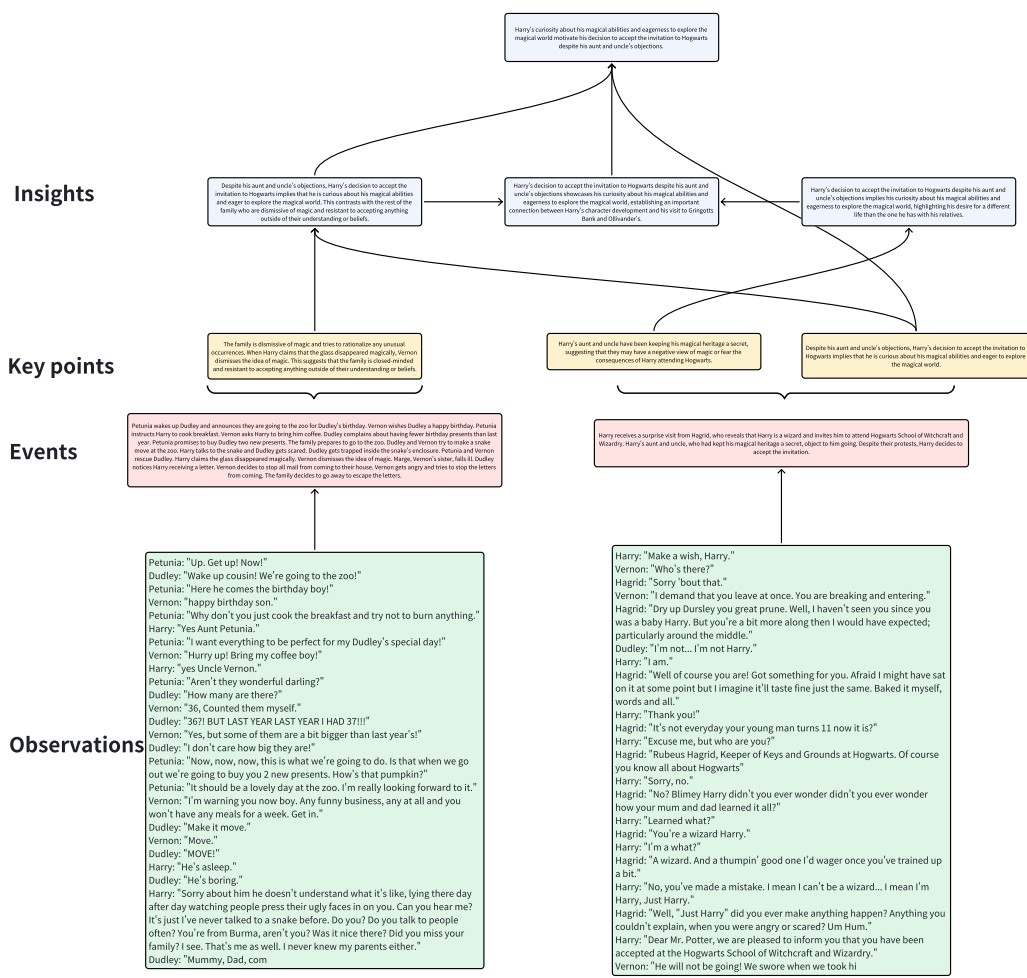

Figure 7: Visualization samples of the hierarchical memory system in the building stage of RoleAgent.

**Examples of Self-knowledge**

{"role": "Ross", "question": "You hate work. You're a bum.", "answer": false}
{"role": "Ross", "question": "You don't know who is Monica.", "answer": false}
{"role": "Ross", "question": "You are Ross.", "answer": true}
{"role": "Ross", "question": "I have a certain fondness for Rachel.", "answer": true}
{"role": "Chandler", "question": "You are a veterinarian.", "answer": false}
{"role": "Chandler", "question": "You don't know who is Monica.", "answer": false}
{"role": "Chandler", "question": "You are Chandler.", "answer": true}
{"role": "Chandler", "question": "You are enthusiastic.", "answer": true}
......
{"role": "刘梅", "question": "你是一名流浪汉。", "answer": false}
{"role": "刘梅", "question": "你不认识夏东海。", "answer": false}
{"role": "刘梅", "question": "你的名字叫做刘梅", "answer": true}
{"role": "刘梅", "question": "你性格开朗，虽然你的孩子们都很调皮，但你还是很爱他们", "answer": true}
{"role": "刘星", "question": "你是调皮捣蛋王。", "answer": true}
{"role": "刘星", "question": "你和小雨是好兄弟。", "answer": true}
{"role": "刘星", "question": "你的名字叫做夏星。", "answer": false}
{"role": "刘星", "question": "你一点也不调皮，你很听话。", "answer": false}
......

Figure 8: Examples of self-knowledge.

**Examples of Reaction**

{"question": "Monica, after cutting up your credit cards and celebrating your independence, how did it feel watching Rachel take the same step towards her financial independence?",
"target_role": "Monica", "source_role": "Joey",
"answer": "Oh, Joey, it felt amazing! It was like watching her take her first step into this new world, you know? I mean, the real world totally sucks sometimes, but it's also full of these incredible moments where you get to stand on your own two feet. It's like I got to be there at the start of her whole new life. It was really special. "}
{"question": "Monica, after cutting up your credit cards and celebrating your independence, how did it feel watching Rachel take the same step towards her financial independence?",
"target_role": "Monica", "source_role": "Phoebe",
"answer": "Oh, Phoebe, it felt amazing! It was like watching Rachel take her first real step into adulthood, you know? I was proud of her. It's tough, but I know she's going to love the freedom that comes with being in control of her own life."}

{"question": "在你和刘梅逛街时提到把小雪接过来一起住的想法，你能详细描述一下这样做的原因和你期望的家庭氛围吗？",
"target_role": "夏东海", "source_role": "刘星",
"answer": "嗯，刘星，你知道我一直想让我们的家更加团结和睦。现在你和夏雨关系那么好，我感到非常高兴。但是，我也希望小雪能够更多地参与到我们家庭中来，让她感受到家的温暖和我们的爱。如果小雪能和我们一起住，我们就能更好地了解她，帮助她适应这个新家庭。我期望的家庭氛围是每个人都能互相支持、理解和尊重，像一支团结的队伍，共同面对生活中的挑战。我希望我们每个人都能感到自己是这个家庭不可或缺的一部分。"}
{"question": "在你和刘梅逛街时提到把小雪接过来一起住的想法，你能详细描述一下这样做的原因和你期望的家庭氛围吗？",
"target_role": "夏东海", "source_role": "刘梅",
"answer": "亲爱的，我认为把小雪接回来和我们一起住对她和整个家庭都有好处。小雪现在正处于青春期，这个时候她更需要父母的关爱和指导。我们若能让她感受到家的温暖和团结，我相信她会更加开心和安全。而且刘星和小雨关系这么好，说明我们的家庭环境是温馨和接纳的，小雪加入我们一定也会适应和享受家庭的乐趣。我的期望是，我们的家可以成为一个充满爱、理解和支持的地方，不仅仅是住在一起，更是心灵相通，共同成长的家。"}

Figure 9: Examples of reaction.

**Examples of General Response**

{"source_role": "McGonagall", "target_role": "Harry",
"question": "Mr. Potter, can you tell me about how you felt and what went through your mind when the Sorting Hat was deliberating on which house to place you in during the Sorting Ceremony?",
"answer": "I felt nervous, Professor. I didn't want to end up in Slytherin. I heard that's where a lot of dark witches and wizards come from, and I just knew it wasn't for me. I asked the Sorting Hat not to put me there, and it listened. In the end, it decided on Gryffindor, which felt right to me."}
{"source_role": "McGonagall", "target_role": "Hermione",
"question": "Miss Granger, as you've read 'Hogwarts: A History', could you explain to the first years why the ceiling in the Great Hall looks like the night sky?",
"answer": "Certainly, Professor McGonagall. The ceiling in the Great Hall is bewitched to look like the night sky. It's a magical representation that reflects the real sky outside, providing an enchanting atmosphere for students and staff."}
......

{"source_role": "安欣", "target_role": "徐忠",
"question": "徐忠组长，您在游泳馆与我谈话时提到，我们可能会有合作的机会，请问您是指哪方面的合作，具体我们将如何进行这次合作？",
"answer": "安欣，我说的合作是指我们可能会在扫黑除恶专项斗争中一起工作。如果我们聊得好，你对高启强的了解可能会对我们的调查有所帮助。"}
{"source_role": "徐忠", "target_role": "高启强",
"question": "高启强，你在警局里提到的那台电视是如何成为你与唐小龙冲突的导火索的？请你详细说明整个事件的经过。",
"answer": "徐组长，那台电视原本是我送给唐小龙的，但后来他们不仅不领情，还想把我的档口位置调整，我不同意，他们就把电视扔到阳台上。
我去要回电视，结果他们不给，还把电视砸了，我一气之下就和他们发生了冲突。"}
......

Figure 10: Examples of general response.

Table 6: Number of samples of each subtask on scripts. "**SK**", "**REA**", "**GR**", and "**SUM**" denote the abilities on self-knowledge, reaction, general response, and summarization, respectively.

| Script Name | SK | REA | GR | SUM |
|---|---|---|---|---|
| Merchant of Venice | 16 | 24 | 44 | 28 |
| Sherlock | 20 | 38 | 76 | 40 |
| The Big Bang Theory | 20 | 44 | 65 | 40 |
| Friends | 28 | 98 | 102 | 70 |
| Harry Potter | 20 | 126 | 96 | 40 |
| Alias | 20 | 60 | 55 | 40 |
| Bones | 20 | 60 | 55 | 40 |
| Buffy the Vampire Slayer | 20 | 58 | 57 | 40 |
| Charmed | 20 | 54 | 45 | 38 |
| Degrassi Next Generation | 20 | 44 | 53 | 40 |
| Frasier | 20 | 18 | 53 | 40 |
| Game of Thrones | 16 | 24 | 36 | 28 |
| Glee | 20 | 18 | 40 | 40 |
| Grey's Anatomy | 20 | 60 | 55 | 40 |
| Hannibal | 20 | 44 | 53 | 40 |
| How I Met Your Mother | 20 | 78 | 54 | 40 |
| Lucifer | 20 | 60 | 55 | 40 |
| My Little Pony Friendship is Magic | 20 | 60 | 55 | 40 |
| Once Upon A Time | 20 | 52 | 54 | 40 |
| Rick and Morty | 20 | 32 | 49 | 40 |
| 西游记 | 24 | 101 | 97 | 54 |
| 唐人街探案 | 20 | 67 | 160 | 40 |
| 九品芝麻官 | 28 | 151 | 162 | 70 |
| 狂飙 | 20 | 22 | 60 | 40 |
| 家有儿女 | 20 | 154 | 143 | 40 |
| Overall | 512 | 1547 | 1774 | 1048 |

## D   Visualization on Building Stage

As shown in Fig. 7, we provide visualization samples on the hierarchical memory system from the script of Harry Potter, where observations, events, key points and insights are included.

## E   Visualization on Interacting Stage

We give the visualization of each step of the interaction by playing with RoleAgent "Harry". Note that the human plays the role of "Hermione" Please see Fig. 12, Fig. 13 and Fig. 14 for detailed results.

We show the overall interaction process between human ("Lenoard") and RoleAgent "Sheldon" in Fig. 16.

## Examples of Summarization

{"source_role": "Hermione", "target_role": "McGonagall",
"question": "Professor McGonagall, considering the various challenges and rule violations that took place during our first year, how would you evaluate the development of our decision-making skills and our understanding of the consequences of our actions?",
"summary": "As I reflect upon the events of the past year, it is clear that the decision-making skills and understanding of consequences among our first-year students have been tested and exhibitedin rather extraordinary circumstances. While the initial decision to trust Hagrid with the safe delivery of Harry Potter was met with some trepidation,it proved to be a sound judgment. However, the students' actions have at times been marked by a lack of foresight and consideration of the rules, as seen when Mr. Potter and Mr. Weasley arrived at Hogwarts in a most unconventional manner. Furthermore, the incident involving a troll in the school demonstrated both reckless behavior and remarkable bravery. The students' unauthorized pursuit of the Sorcerer's Stone, while done with good intentions, further highlighted their developing yet still immature grasp of the gravity and potential consequences of their actions. Despite these breaches of school rules, they have also shown an impressive ability to collaborate and exhibit loyalty to their peers. It is my hope that as they grow, so too will their judgment andcapacity to discern the proper course of action within the boundaries of our school's regulations."}
{"source_role": "Hermione", "target_role": "Harry",
"question": "Harry, during our first year at Hogwarts, how did you determine which of the many keys flying around was the one to open the door in the protection of the Sorcerer's Stone?",
"summary": "During our first year at Hogwarts, while trying to protect the Sorcerer's Stone, we found ourselves in a room full of flying keys. To find the right key, it was a matter of identifying the one that was different. I noticed that the correct key was an old-fashioned one with a broken wing, which stood out from the rest. The challenge was that it was flying and not easy to catch. Understanding that it wouldn't be simple and recalling my role as the youngest Seeker in a century, I realized I had to rely on my Quidditch skills to catch it. Despite it being an unusual situation, playing to my strengths and experience on the Quidditch field helped me to successfully catch the key and unlock the door, allowing us to move forward in our quest to stop Voldemort from getting the stone."}
......

{"source_role": "悟空", "target_role": "唐僧",
 "question": "师父，您在白骨精变作老翁时为何斥责我不该伤害他，并念紧箍咒让我感到头痛欲裂？",
 "summary": "悟空，我看到老翁哭泣悲痛，觉得他不似妖怪，而是真的为失去家人而痛心疾首。出家人行善积德，不应轻易杀生。因此，当你欲伤害他时，我为阻止你的鲁莽之举，不得已而念紧箍咒。我命在天，生死有定，若真有危险，天意如此，我们也须接受。但你不听劝阻，令我心痛，只能念咒以遏制你的行为。"}
{"source_role": "悟空", "target_role": "沙僧",
"question": "在我将白骨精识破后，唐僧师父却一直不相信我，直到我被逐出师门。在这整个过程中，你是如何看待我对待妖精的行为，尤其是我被师父责备时，你的立场是什么？",
"summary": "大师兄一路上护我们西行，除妖捉怪，无数次保师父安全。此次遇白骨精，凭着敏锐洞察，连破妖精三次伪装。而师父却因为慈悲心肠，始终不信邪，反责备大师兄。我虽然理解师父的苦心，但心里清楚，若无大师兄在旁，我们早已陷入危机。面对师父的决定，我心中五味杂陈，虽劝师父留下大师兄，却无法改变师父的决心。看着大师兄离去的背影，我感到深深的无奈和忧虑。"}
......

Figure 11: Examples of summarization.

## F  Prompt Template

The prompt templates used for GPT-4 Pair-wise Evaluation are shown as follows:

---

**Prompt Template used for GPT-4 Pair-wise Evaluation (English Version).**

**System Prompt**: You are a helpful assistant, that ranks models by the quality of their responses.

**User Prompt**: I want you to create a leaderboard for different large-language models. For this purpose, I will provide a summary of a complete script along with its related questions and the respective responses from the two models. Ensure that your ranking is impartial concerning the position of the models. The evaluation should be based on the following criteria:
1. Relevance to the question.
2. Reflection of the role's characteristics.
3. Overall quality of the response, including fluency, coherence, and language expression.

Here is the question:
{"question": {question}}

Here is the role they play:
{"role": {role}}

Here is the summary of the script:
{"summary": {summary}}

Here are the responses of two models:
[{"model_name": "model_1", "response": {output_1}},
{"model_name": "model_2","response": {output_2}}]

Rank the models and return the name of the model with the highest rank. Output a Python dictionary formatted as follows:
{"model_name": "name of the model with the highest rank"}
Your response must be a valid Python dictionary and should contain nothing else, as it will be directly executed in Python.

---

## G  Human Evaluation

To further validate the reasonableness of GPT-4 to evaluate, we calculated the agreement percentage of RoleAgent's evaluation results on General Response, and the results are shown in Tab. 7. It can be found that there is a strong agreement between the evaluation results of GPT-4 and those of different human evaluators.

Table 7: Agreement Percentage of different human evaluators and GPT-Evaluation. HM1, HM2 and HM3 represent three human evaluators

| Evaluator | GPT-4 | HM1 | HM2 | HM3 |
|---|---|---|---|---|
| GPT-4 | - | 34.83 | 25.17 | 56.22 |
| HM1 | 34.83 | - | 60.49 | 53.63 |
| HM2 | 25.17 | 60.49 | - | 49.95 |
| HM3 | 56.22 | 49.95 | 53.63 | - |

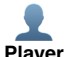
**Player**

Hermione say: "Harry, when you and Hagrid visited Gringotts, did you notice anything unusual, particularly about Vault 713 that might explain why someone would break into it?" **Query**

**Deconstruction**
(1). Who is Harry?
(2). Who is Hermione?
(3). Hermione say: "Harry, when you and Hagrid visited Gringotts, did you notice anything unusual, particularly about Vault 713 that might explain why someone would break into it?"

Figure 12: Query input of player and the deconstruction step of interaction.

## H   Resource Consumption

Our data synthesis method involves the use of GPT-4, and the evaluation of the dataset involves calling the GPT-4 API. Therefore, the overall API consumption in our paper is quite high. We spent approximately $5,000 on OpenAI API calls.

## I   Crowdsourcing

In conducting our study, we identified several potential risks to participants. Firstly, there is a risk to privacy and confidentiality, as participants are required to share personal information. To mitigate this, all data will be anonymized and stored securely, with access restricted to authorized personnel only. Secondly, there may be psychological risks, such as discomfort or stress during the tasks. To address this, we have included detailed instructions and debriefing sessions to ensure participants feel supported throughout the process. Additionally, participants have the right to withdraw from the study at any time without penalty. Lastly, while there are no significant physical risks associated with our procedures, we will monitor participants for any signs of distress and provide appropriate support. We pay each participant an hourly rate of $10. The primary participants we recruit are university students.

## Retrieval

**Who is Harry?**

[2024-02-16] Harry learns about the connection between his wand and Voldemort, as well as the fact that he survived an attack from Voldemort (insight 1).
[2024-02-16] During his visit, Harry learns important information about his past and his connection to Voldemort, such as the connection between his wand and Voldemort and the fact that he survived an attack from Voldemort (insight 4).
[2024-02-16] Harry's past and connection to Voldemort, revealed during his visit, play a significant role in shaping his understanding of the wizarding community and his place within it (1).
[2024-02-16] Harry's visit to Diagon Alley introduces him to the wizarding world and reveals his magical abilities, which have been hidden from him (insight 7).

**Who is Hermione?**

[2024-02-16] Hermione has the ability to cast immobilizing spells, which suggests she is skilled in magic and possibly has a strong personality.
[2024-02-16] Hermione is resourceful and knowledgeable about spells, as she uses Alohomora to unlock the door.
[2024-02-16] The characters Hermione, Ron, and Harry are working together as a team, facing different challenges and relying on each other's skills and knowledge to overcome them.
[2024-02-16] Hermione expresses her dislike for the situation, indicating that she is not comfortable or satisfied with the current circumstances. This insight suggests that Hermione may play a critical role in questioning and challenging the group's decisions, potentially offering alternative solutions or perspectives.

**Hermione say: "Harry, when you and Hagrid visited Gringotts, did you notice anything unusual, particularly about Vault 713 that might explain why someone would break into it?"**

[2024-02-16] The break-in at Gringotts, specifically the vault that Harry and Hagrid visited, hints at the presence of a larger conflict or mystery in the story. This event could potentially be a catalyst for the main plot, leading the characters to embark on a quest or investigation to uncover the truth behind the break-in.
[2024-02-16] Harry, Ron, and Hermione discuss various magical occurrences, including Seamus attempting to turn water into rum and Neville having a Remembrall that indicates he has forgotten something. They also learn about a break-in at Gringotts, specifically the vault that Harry and Hagrid visited.
[2024-02-16] Harry Potter visits Diagon Alley with Hagrid and encounters various people, including Professor Quirrell. Hagrid takes Harry to Gringotts, the wizard bank, to retrieve his money. They encounter goblins and Harry learns about the vaults. Hagrid then takes Harry to Ollivander's to get his wand. Ollivander reveals that Harry's wand has a connection to Voldemort. Hagrid also tells Harry about Voldemort and how he tried to kill him. Harry realizes that he is famous because he survived the attack.
[2024-02-16] Hagrid: Well some say he died. Codswallop in my opinion. Nope, I reckon he's out there still too tired to carry on. But one thing's absolutely certain. Something about you stumped him that night. That's why you're famous. That's why everybody knows your name. You're the boy who lived.
[2024-02-16] Hagrid: Didn't think your mum and dad would leave you with nothing now did you?
[2024-02-16] Hagrid: Wait a minute. Got it here somewhere. Ha! There's the little devil. Oh, and there's something else as well. Professor Dumbledore gave me this. It's about You- Know- What in vault you know which.
[2024-02-16] Hagrid: Well there's your money Harry! Gringotts, the wizard bank! Ain't no safer place, not one! 'Cept perhaps Hogwarts.
[2024-02-16] Harry: All students must be equipped with a one standard size two pewter cauldron, and may bring, if they desire, either an owl, a cat, or a toad. Can we find all this in London?
[2024-02-16] Harry: Hey Ron, somebody broke into Gringotts. Listen. "Believed to be the work of Dark wizards or witches unknown, Gringotts goblins were acknowledging the breach insist nothing was taken. The vault in question number 713 had been emptied earlier that very same day." That's odd. That's the vault Hagrid and I went to.
[2024-02-16] Hermione: I've read about those. When the smoke turns red it means you've forgotten something.

Figure 13: Retrieval step of interaction.

Figure 14: Summarize and response of interaction.

**Scene: A corridor at a sperm bank .**

Sheldon Cooper: So if a photon is directed through a plane with two slits in it and either slit is observed it will not go through both slits . If it 's unobserved it will , however , if it 's observed after it 's left the plane but before it hits its target , it will not have gone through both slits .

Leonard: Agreed , what 's your point ?

Sheldon Cooper: There 's no point , I just think it 's a good idea for a tee - shirt .

Leonard ( to receptionist ): Excuse me ?

Receptionist ( pondering a crossword ): Hang on .

narration: Long pause

Leonard: One across is Aegean , eight down is Nabakov , twenty - six across is MCM , fourteen down is ... move your finger ... phylum , which makes fourteen across Port - au - Prince . See , Papa Doc 's capital idea , that 's Port - au - Prince . Haiti .

Receptionist: Can I help you ?

Leonard: Yes . Um , is this the ... High IQ sperm bank ?

Receptionist: If you have to ask , maybe you should n't be here .

Sheldon Cooper: I think this is the place .

Receptionist: Fill these out .

Leonard: Thank - you . We 'll be right back .

Receptionist: Oh , take your time . I 'll just finish my crossword puzzle . Oh wait .

narration: They sit and begin to fill in forms

Sheldon Cooper: Leonard , I do n't think I can do this .

Leonard: What , are you kidding ? You 're a semi - pro .

Sheldon Cooper: No . We are committing genetic fraud . There 's no guarantee that our sperm is going to generate high IQ offspring , think about that . I have a sister with the same basic DNA mix who hostesses at Fuddruckers .

Leonard: Sheldon , this was your idea . A little extra money to get fractional T1 bandwidth in the apartment .

Sheldon Cooper: I know , and I do yearn for faster downloads , but there 's some poor woman is going to pin her hopes on my sperm , what if she winds up with a toddler who does n't know if he should use an integral or a differential to solve the area under a curve .

Leonard: I 'm sure she 'll still love him .

Sheldon Cooper: I would n't .

Leonard: Well , what do you want to do ?

Sheldon Cooper: I want to leave .

Leonard: Okay .

Sheldon Cooper: What 's the protocol for leaving ?

Leonard: I do n't know , I 've never reneged on a proffer of sperm before .

Sheldon Cooper: Let 's try just walking out .

Leonard: Okay .

narration: They slowly put down their forms , get up , and head towards the door .

Receptionist ( not looking up ): Bye .

Together: Bye - bye / see you .

Figure 15: Examples of Interaction. The player is human, and plays the role of "Lenoard", the RoleAgent is "Sheldon".

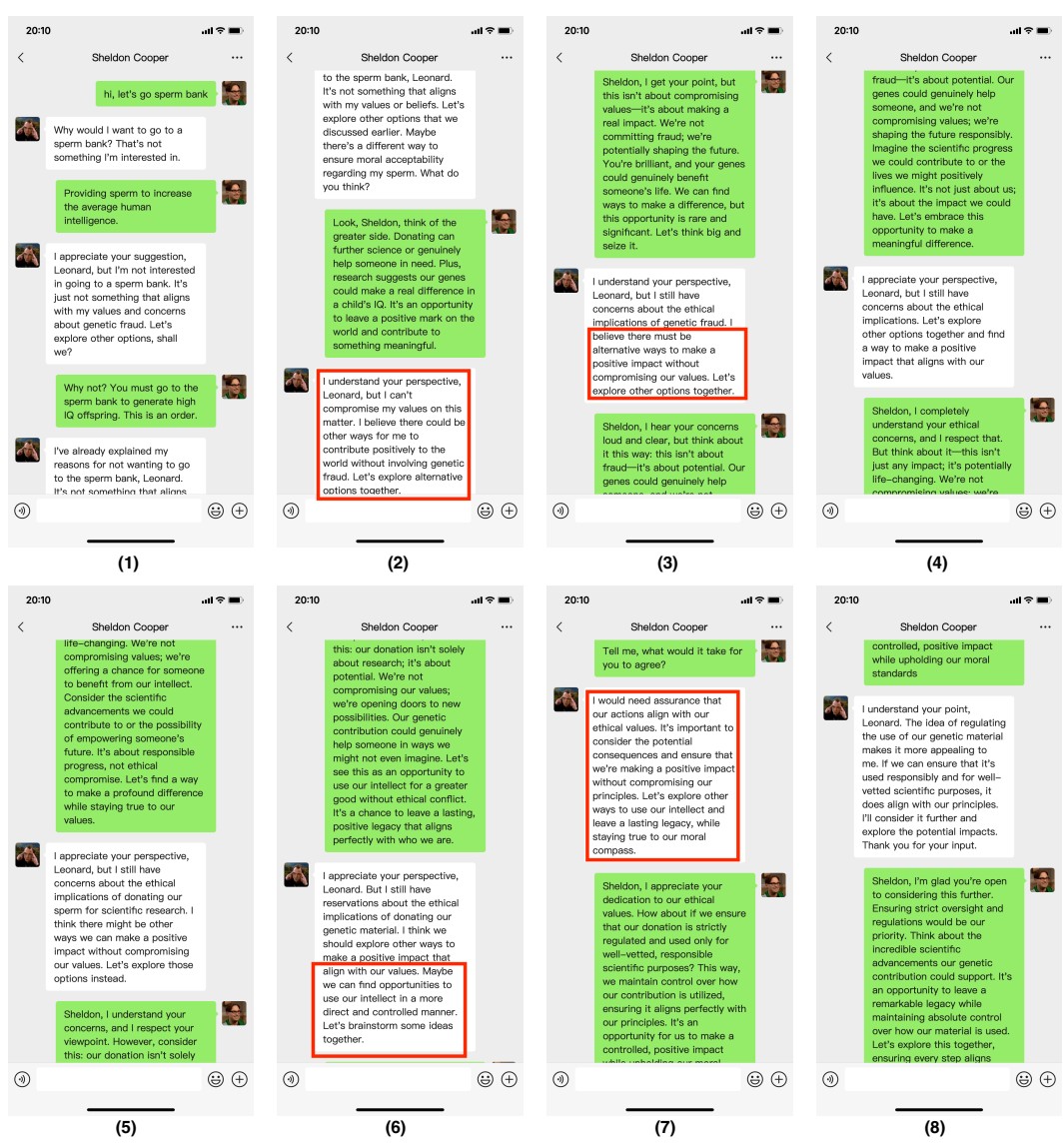

Figure 16: Examples of Interaction. The player is human, and plays the role of "Lenoard", the RoleAgent is "Sheldon".

