# OpenReview forum: "RoleAgent: Building, Interacting, and Benchmarking High-quality Role-Playing Agents from Scripts"
_NeurIPS.cc/2024/Datasets_and_Benchmarks_Track — NeurIPS 2024 Track Datasets and Benchmarks Poster_

### Official Review · Reviewer_rkjx · 2024-06-22
**Good work**

**Rating:** 8
**Confidence:** 4
**Clarity:** Yes

**Review:**

Quality: The paper presents a well-structured and innovative approach to generating role-playing agents from raw scripts. The methodology is robust, with clear stages and a detailed hierarchical memory system that enhances agent interactions. The experiments are comprehensive, covering multiple benchmarks and providing a thorough comparison with existing methods.

Clarity: The paper is clearly written, with each section logically following the previous one. However, some parts, particularly the hierarchical memory construction and the dynamic importance scoring mechanism, could benefit from additional clarification and examples to aid understanding.

Originality: The framework's originality lies in its ability to automate the creation of agents from raw scripts without extensive human intervention. This is a significant advancement over existing methods that rely heavily on manual annotation.

Significance: The proposed framework has the potential to impact various applications, from immersive environments to interactive learning tools. The comprehensive evaluation benchmark further strengthens the paper's contributions by providing a standardized way to assess agent quality.

Pros:

- Innovative approach to generating role-playing agents.
- Robust hierarchical memory system.
- Comprehensive evaluation benchmark.
- Extensive experimental validation.

Cons:

- Some methodological details lack clarity.
- Evaluation focuses on a limited set of scripts, raising questions about generalizability.

**Strengths:**

- Significance of Contribution: The framework addresses a crucial limitation in current generative agent models by eliminating the need for manual annotation, thereby enhancing scalability.
- Relevance: Highly relevant to the broader research community focused on NLP, AI, and interactive systems.
- Quality of Research: The hierarchical memory system and the dynamic interaction mechanism are well-developed and innovative.

**Additional Feedback:**

- Consider integrating a real-time feedback mechanism to dynamically adjust agent behavior based on user interactions.
- Explore the potential of combining RoleAgent with reinforcement learning techniques to further enhance agent believability and interaction quality.
- Provide a more detailed breakdown of the computational resources required for both the building and interacting stages.

**Correctness:**

The claims made in the submission appear to be correct based on the provided experimental results. The evaluation methods and experiment design are appropriate and seem to perform reliably

**Documentation:**

The documentation for the datasets and benchmarks is detailed and sufficient to support reproducibility.

**Limitations:**

The authors have acknowledged the limitations related to the costs associated with API usage and GPU consumption. However, they should provide more details on the computational efficiency of their approach. Additionally, the potential biases introduced by the hierarchical memory system should be addressed more thoroughly.

**Opportunities For Improvement:**

- Methodological Clarity: Provide more detailed explanations and examples, particularly for the hierarchical memory construction and dynamic importance scoring mechanisms.
- Evaluation Diversity: Include a more diverse set of scripts to test the generalizability of the framework across different genres and contexts.

**Relation To Prior Work:**

Yes

**Summary And Contributions:**

The paper introduces the RoleAgent framework, which aims to create scalable, high-quality role-playing agents from raw scripts without the need for extensive human annotation. This framework consists of two main stages: building and interacting. The building stage involves a hierarchical memory system to extract and summarize agent profiles from scripts, while the interacting stage includes a novel four-step interaction mechanism. The authors also propose a comprehensive evaluation benchmark, RoleAgentBench, to assess the effectiveness of RoleAgent. Extensive experiments demonstrate the framework's advantages over existing methods.

---

> ### Author Rebuttal · Authors · 2024-08-17
>
> Thank you for your constructive comments. Here are some explanations to address your questions.
>
> **Q1. Clarity on  methodological details (i.e., the hierarchical memory construction and the dynamic importance scoring mechanism).**
>
> **A1**. In Fig. 6 of the Appendix.C, we take the script of Harry Potter as an example to provide the visualization samples on the hierarchical memory system, where observations, events, key points, and insights are included. Specifically, as discussed in Line 110-139, from the parsed observations, we first extract the events and then obtain the key points for these events. After that, we prompt the LLMs to reflect the insights based on these key points and the corresponding events. Finally, observations are progressively abstracted into events, key points, and insights, which can greatly reduce the redundancy of raw observations.
> For the dynamic importance score mechanism, the dynamic importance score is the frequency with which memory is retrieved throughout the interaction stage, denoting the relevance between the memory and the interaction queries. In our new version, we will update an interaction video demo to show the process of score changes for better illustration.
> We will carefully rewrite these sections for better clarity in our new version.
>
> **Q2. Evaluation on a limited set of scripts.**
>
> **A2**. In the main paper, we focus on 20 English and 5 Chinese scripts, and it is important to cover more scripts to demonstrate the generalizability of our RoleAgent.  Therefore, we refer to the online film database IMDb [https://www.imdb.com/], which divides mainstream films into 22 categories  Action, Adventure, Animation, Biography, Comedy, Crime, Documentary, Drama, Family, Fantasy,  History, Horror, Music, Musical, Mystery, Romance, Sci-Fi, Short, Sport, Thriller, War, Western. After checking the categories of the existing 20 scripts, we observe that five categories (i.e., Biography, Documentary, History, Sport, Western) have not been covered in our scripts. Thus, we additionally select five scripts (i.e., Saints & Strangers, Friday Night Lights, Hell on Wheels, Mars, The Grand Tour) to cover corresponding scripts. Then, we conduct the building process and prepare the evaluation datasets for these five scripts on the four subtasks, where the evaluation results are provided in the following table. We will add the discussion and supplement these results in our new version.
>
> | Models      | Self-Knowledge (**Acc.**) | Reaction (Acc. ) | General Response (WR-G) | Summarization (ED) | Summarization (ER) | Summarization (ER/ED) | Summarization (WR-G) |
> | ----------- | ------------------------- | ---------------- | ----------------------- | ------------------ | ------------------ | --------------------- | -------------------- |
> | Qwen-Max    | 87.2                      | 84.2             | 47.5                    | 6.4                | 26.0               | 4.06                  | 69.4                 |
> | GPT-3.5     | 83.5                      | 68.0             | 40.1                    | 13.7               | 42.5               | 3.10                  | 58.4                 |
> | Qwen1.5-14B | 82.6                      | 64.7             | 37.8                    | 7.9                | 25.5               | 3.23                  | 50.2                 |
> | LLaMA3-8B   | 83.4                      | 63.2             | 39.2                    | 8.2                | 26.4               | 3.22                  | 45.3                 |
>
> **Q3. Details on the computational efficiency.**
>
> **A3**. For closed-source models (e.g., GPT-4 and GPT-3.5),  in the building stage, we use about 184k tokens for each script on average, and in the interacting stage, we use about 1k tokens for each question on average.
> For open-sourced models, to support the inference on building and interaction stages, we mainly use one  A100 (80G) GPU for Qwen1.5-14B, Mistral-7B, LLaMA3-8B, ChatGLM3-6B, and we use two A100 (80G) GPUs for Yi-34B. Note that token consumption is the same as closed-source models.
> We will add this detail in the new version.
>
> **Q4. Potential biases introduced by the hierarchical memory system.**
>
> **A4**. We acknowledge the limitation that the hierarchical memory system may lead to potential bias, as the memory is generated by LLMs and the abstraction process may lose important contexts in the raw scripts. As discussed in **G.Q1** of the **General Response**, we conduct a detailed evaluation of the memory system and observe that the automatically generated hierarchical memories are well-performed on both the subjective and objective evaluation metrics. In the future, we will continue to investigate how to reduce the potential bias introduced by the hierarchical memory system.

---

> ### Author Rebuttal · Authors · 2024-08-30
>
> **Q5.Integrate a real-time feedback mechanism.**
>
> **A5**. As discussed in the ethics statement of the supplementary material, we also propose that a comprehensive feedback mechanism can enable users to voice ethical concerns and report troubling results to support ongoing refinement and responsibility.
> Besides, implicit feedback exists in our RoleAgent framework. Specifically, in the interaction stage, we dynamically extract new hierarchical memories (i.e., events, key points, and insights) based on the current interaction record, where the implicit abstraction and reflection are applied on the interaction process. In other words, the reflection memories (e.g., insights) can represent the implicit feedback on the current interaction, which can improve the interaction quality. Besides, there are many works [R1, R2, R3] related to the explicit feedback mechanism to improve LLM performance, which can improve agent performance greatly. For example, the Self-Refine proposes to improve initial outputs from LLMs through iterative feedback and refinement based on prompt engineering. However, it should be mentioned that the feedback mechanism introduces additional API or computation costs. In the future, we will continue to investigate how to build an effective feedback mechanism.
>
> [R1] Self-Refine: Iterative Refinement with Self-Feedback
>
> [R2] Agent-Pro: Learning to Evolve via Policy-Level Reflection and Optimization
>
> [R3] The ART of LLM Refinement: Ask, Refine, and Trust
>
>
> **Q6. Combin RoleAgent with reinforcement learning.**
>
> **A6**. The reinforcement learning [R1, R2, R3] has been widely used to make the LLM-based agents align with human preferences and improve the interaction quality, where we need to prepare high-quality positive and negative pairs annotated by humans or well-performed LLMs (e.g., GPT-4). In future work, we will explore building an automatic annotation system to produce high-quality positive and negative pairs and using these datasets to improve the agent believability and interaction quality.
>
> [R1] Re-ReST: Reflection-Reinforced Self-Training for Language Agents
>
> [R2] Training language models to follow instructions with human feedback
>
> [R3] Direct Preference Optimization: Your Language Model is Secretly a Reward Model

---

### Official Review · Reviewer_M7P2 · 2024-07-06
**Review on paper 908**

**Rating:** 7
**Confidence:** 4
**Correctness:** The claims made in the submission are…
**Clarity:** Yes.

**Review:**

Please see below "Strengths" and "For Improving".

**Strengths:**

1. The proposed RoleAgent includes all the essential parts to construct an agent, with some designs on the role-play task. For the perception part, the query deconstruction step could reflect on (1) the agent itself, (2) the player, and (3) the provided query. It could make the perception more comprehensive. For the memory part, it designs the retrieval process and summarization process, which empowers the agent more basis to conduct LLM inference. For the action part, an assembled mechanism can also improve a better response intuitively.
2. I think the benchmark is great, and the three proposed evaluation perspectives seem interesting. The self-knowledge can verify the role-play effectiveness from an inside manner, while the reaction and general response could do this from an outside manner.
3. This paper provides details of the agent design, as well as the evaluation benchmark with results. The demonstration is clear to make readers get the points easily.

**Additional Feedback:**

Please see the above comments.

**Documentation:**

Yes, there are sufficient details to support reproducibility.

**Ethics:**

No.

**Limitations:**

The authors have discussed the limitations in Appendix.

**Opportunities For Improvement:**

1. Actually, I am greatly curious about how GLM-4 performs in this setting, because as I know, it has great ability on the role-play task. You may use the open-source GLM-4-9B or GLM-4 API.
2. For the evaluation of memory, besides summarization, I think how the agent could retrieve proper memory is also significant. Whether the agent could find the correct memory is important to express their specific features based on the query. You may refer to a survey on the memory of LLM-based agents [1].
3. Recently, many LLMs could do well in long-context (e.g. 128k in GLM-4 series), what about the agent performance if the memory changes ranking correlation score into LLM retrieval?
If the author could address my concerns in the rebuttal period, I am willing to improve my rating.

Reference:

[1] Zhang, Z., Bo, X., Ma, C., Li, R., Chen, X., Dai, Q., ... & Wen, J. R. (2024). A survey on the memory mechanism of large language model based agents. arXiv preprint arXiv:2404.13501.

**Relation To Prior Work:**

Yes.

**Summary And Contributions:**

This paper proposes a new framework of LLM-based agents for the role-play task called RoleAgent, which could generate high-quality roles from raw scripts. This framework includes the building stage and the interacting stage. During the building stage, it proposes a hierarchical memory system to empower the agent more reliable characteristics by extracting and summarizing the obtained information. During the interacting stage, it uses four steps to take actions based on observations, including query deconstruction, memory retrieval, memory summarization, and response generation. To evaluate the RoleAgent, this paper also constructs a benchmark called RoleAgentBench with both Chinese and English scripts. The results show the effectiveness of the proposed methods.

---

> ### Author Rebuttal · Authors · 2024-08-17
>
> Thanks for your insightful comments. We will address your concerns shown below in detail.
>
> **Q1. Results of GLM-4-9B and GLM-4 API.**
>
> **A1**. The results of GLM-4-9B and GLM-4 API are shown in the following table. On the English subset, we observe that the GLM-4 API results are close to the GPT-4. On the Chinese subset, we observe that the results of GLM-4 API are comparable with GPT-4 and even better than GPT-4 on several evaluation subtasks (i.e., Self-Knowledge, Reaction).  Moreover, for GLM-4-9B, we also achieve competitive results in both subsets. Specifically, the results of GLM-4-9B are comparable with the results of Yi-34B and GPT-3.5, which shows the effectiveness of GLM-4-9B on role-playing. We will revise the above discussion in our new version.
>
> | English subset       | Self-Knowledge (Acc.) | Reaction (Acc. ) | General Response (WR-G) | Summarization (ED) | Summarization (ER) | Summarization (ER/ED) | Summarization (WR-G) |
> | -------------------- | --------------------- | ---------------- | ----------------------- | ------------------ | ------------------ | --------------------- | -------------------- |
> | RoleAgent (GPT-4)    | 94.7                  | 88.9             | 48.6                    | 8.9                | 39.8               | 4.47                  | 75.8                 |
> | RoleAgent (GLM-4-9B) | 90.8                  | 74.2             | 34.9                    | 11.5               | 37.4               | 3.25                  | 52.3                 |
> | RoleAgent(GLM-4 API) | 93.3                  | 86.2             | 48.3                    | 9.9                | 44.7               | 4.52                  | 72.5                 |
>
> | **Chinese subset**   | Self-Knowledge (**Acc.**) | Reaction (Acc. ) | General Response (WR-G) | Summarization (ED) | Summarization (ER) | Summarization (ER/ED) | Summarization (WR-G) |
> | -------------------- | ------------------------- | ---------------- | ----------------------- | ------------------ | ------------------ | --------------------- | -------------------- |
> | RoleAgent (GPT-4)    | 92.5                      | 73.9             | 44.8                    | 10.0               | 45.0               | 4.50                  | 75.1                 |
> | RoleAgent (GLM-4-9B) | 92.4                      | 68.1             | 31.1                    | 11.4               | 43.8               | 3.84                  | 64.5                 |
> | RoleAgent(GLM-4 API) | 92.9                      | 77.3             | 45.9                    | 9.2                | 44.9               | 4.88                  | 74.8                 |
>
> **Q2. Evaluation on the memory system.**
>
> **A2**. Please see **G.Q1** of the **General Response**.
>
>
>
> **Q3. For long-context LLMs (e.g. 128k in GLM-4 series), what about agent performance if the memory changes ranking correlation score into LLM retrieval?**
>
> **A3**. The effectiveness of RAG systems and long-context LLMs on long-context tasks has been investigated in many works [R1, R2]. We take the GLM-4 API as an example to compare the results of our default RoleAgent and the RoleAgent (LC) in the English subset. Specifically, for   RoleAgent (LC), we directly input the memories into the context window as much as possible. Note that we set the maximum context length as 100k. Besides, for  RoleAgent (LC), we propose three variants (i.e., RoleAgent (LC, random), RoleAgent (LC, Ascend), RoleAgent (LC, descend)). For RoleAgent (LC, random), we just randomly the orders of memories. For  RoleAgent (LC, Ascend) and RoleAgent (LC, descend), we input the memories based on the similarity scores using ascending and descending orders, respectively. In the following table, we observe that different variants of the long-context LLM (i.e., GLM-4) achieve inferior results when compared with the results of our RoleAgent based on memory retrieval.
>
> We assume the reasons are as follows:
>
> First, the extracted memories are often abstract and informative and these tasks in our RoleAgentBench often need to reason on multiple memories for obtaining the final results. However, the existing long-context LLMs usually achieve competitive performance on the retrieval task (e.g., Needle in the Haystack), where the reasoning abilities on the multiple cues from different positions are still challenging. Second, the memory retrieval process in our RoleAgent reduces the redundancy of irrelevant memories and the reasoning abilities of existing LLMs in relatively short contexts are also better than the abilities of long contexts, which makes our RoleAgent achieve better performance on our RoleAgentBench.
>
> We will revise this discussion in our new version.
>
> | **English subset**      | Self-Knowledge (**Acc.**) | Reaction (Acc. ) | General Response (WR-G) | Summarization (ED) | Summarization (ER) | Summarization (ER/ED) | Summarization (WR-G) |
> | ----------------------- | ------------------------- | ---------------- | ----------------------- | ------------------ | ------------------ | --------------------- | -------------------- |
> | RoleAgent               | 93.3                      | 86.2             | 48.3                    | 9.9                | 44.7               | 4.52                  | 72.5                 |
> | RoleAgent (LC, random)  | 87.5                      | 78.4             | 43.4                    | 11.2               | 37.2               | 3.32                  | 66.4                 |
> | RoleAgent (LC, Ascend)  | 91.0                      | 82.1             | 45.3                    | 10.4               | 42.3               | 4.07                  | 65.9                 |
> | RoleAgent (LC, descend) | 89.4                      | 80.3             | 44.7                    | 12.5               | 43.5               | 3.48                  | 63.5                 |
>
> [R1] Retrieval Augmented Generation or Long-Context LLMs? A Comprehensive Study and Hybrid Approach
>
> [R2] Can Long-Context Language Models Subsume Retrieval, RAG, SQL, and More?

---

> > ### Comment · Reviewer_M7P2 · 2024-08-23
> > **Raise Rating**
> >
> > Thanks for the detailed rebuttal by the authors. I would like to raise my rating to 7.

---

### Official Review · Reviewer_HBPU · 2024-07-26
**Accept for novel idea and benchmarking**

**Rating:** 7
**Confidence:** 4
**Correctness:** Correct
**Clarity:** The paper is well written

**Review:**

See Summary And Contributions/Strengths/Weakness

**Strengths:**

1. A pipeline to extract role behavior and personality from script.
2. A new benchmark from current scripts to evaluate personalized agents.

**Additional Feedback:**

N/A

**Documentation:**

N/A

**Limitations:**

See Opportunities For Improvement*

**Opportunities For Improvement:**

1. No detailed illustration on script parsing step, which can be important to incorporate new scripts.

2. More detail regarding experiments is needed. Are different LLMs used for extracting personality or only for question answering?

**Relation To Prior Work:**

Yes

**Summary And Contributions:**

This paper proposes RoleAgent, a framework for extracting character behavior and personality from scripts. Based on the scripts, RoleAgent is able to extract the relationships between characters. Several new methods are proposed to empower this process including memory aggregation, distillation and reflection to obtain hierarchical memory to store the related information contained within scripts. Based on the framework, authors also propose RoleAgentBench from 5 Chinese and 20 English scripts. It can be evaluated with 4 subtasks with human annotated results.

---

> ### Author Rebuttal · Authors · 2024-08-17
>
> Thanks for your careful reading and constructive suggestions. We will address your concerns shown below in detail.
>
> **Q1. Detailed illustration on script parsing.**
>
> **A1**. Thanks for your insightful suggestions. As discussed in Line 98-109 of the main paper, we first cut the raw script into scenes arranged in chronological order. Then, we determine the relevant role and generate role profiles. We have uploaded a figure file based on the script ''Sherlock'' for better illustration.
>
> We will revise the corresponding section and add the figure in our new version.
>
>
>
>
>
> **Q2. Are different LLMs used for extracting personality or only for question answering?**
>
> **A2**. Thanks for your suggestions. We need to clarify that different LLMs are used for both extracting personality in the building phase and answering questions in the interaction phase, which aims to evaluate the overall abilities (e.g., information extraction, summarization and etc.)  in both stages for different LLMs.
>
> We will add this detail in our new version.

---

### Official Review · Reviewer_dVmX · 2024-07-29
**Review for RoleAgent**

**Rating:** 6
**Confidence:** 4
**Correctness:** Yes
**Clarity:** Yes. This paper is well organized and…

**Review:**

- This paper has novel ideas and comprehensive data collection framework for building role agents from scripts.

- The benchmark for evaluating the role agent add critical points to the LLM simulation areas.

- The memory of agent is parsed from three-tire steps, which is able to handle noise and be more structured. However, since there is no human verification in the process, the memory parsing process may not be convincing enough to reflect the actual role behavior or structured knowledge.

- The memory retrieval process involves embedding the query and knowledge, which is however not analyzed comprehensively to compare the performance of different retrievors.

- The response is directly generated via concating subconscious, interactions and relevant content. Though it contains enough informative information, however, some personality or response characteristics are not well revealed, which may not degrade the quality of response.

**Strengths:**

- Novel ideas about the data collection pipeline
- A comprehensive work for role agent data collection, including a good benchmark
- The experiments and data is of good quality for both evaluation and future training of models.

**Additional Feedback:**

NA

**Documentation:**

Dataset is released.

**Ethics:**

The dataset are all from scripts. Though role play may very likely to raise ethical concerns, this paper should be good.

**Limitations:**

Authors mention ethical concerns and computing resources. Other limitation:

- Only scripts may not reflect all the real human roles.

- Restrictions in social or law concerns.

**Opportunities For Improvement:**

- More human verification about the process, especially those involving LLM retrieval and summarization.

- More investigation regarding the personality of the role agent.

- No running code available online.

**Relation To Prior Work:**

Yes

**Summary And Contributions:**

This paper propose a flexible RoleAgent framework and collects the role agent profiles and memories from scripts. Additionally, a this paper introduces a comprehensive role agent benchmark covering 20 English and 5 Chinese scripts. Overall, this paper is novel and interesting. And the work is comprehensive and motivative.

---

> ### Author Rebuttal · Authors · 2024-08-17
>
> Thanks for your careful reading and constructive suggestions. We will address your concerns shown below in detail.
>
> **Q1. Evaluation (human verification) on the memory parsing process.**
>
> **A1**. Please See **G.Q1** in the **General Response**.
>
> **Q2. Performance of different retrievers.**
>
> **A2**. In our main paper, we use OpenAI's text-embedding-ada-002 by default to compute the cosine similarity. To demonstrate the effectiveness of different retrievers on the English subset of our RoleAgentBench, we additionally use two alternative embedding models (i.e., bge-base-en  [https://huggingface.co/BAAI/bge-base-en], GritLM-7B [https://huggingface.co/GritLM/GritLM-7B]) to extract the embedding vectors of memories for retrieving on the English subset of RoleAgentBench based on GPT-3.5. The results are as follows:
>
> | English subset     | Self-Knowledge (Acc.) | Reaction (Acc. ) | General Response (WR-G) | Summarization (ED) | Summarization (ER) | Summarization (ER/ED) | Summarization (WR-G) |
> | ------------------ | --------------------- | ---------------- | ----------------------- | ------------------ | ------------------ | --------------------- | -------------------- |
> | RoleAgent          | 87.6                  | 76.7             | 37.6                    | 11.6               | 36.8               | 3.17                  | 56.7                 |
> | RoleAgent (bge-base-en)    | 88.8                  | 78.4             | 38.4                    | 11.8               | 38.3               | 3.25                  | 63.2                 |
> | RoleAgent (GritLM) | 89.6                  | 81.2             | 42.1                    | 10.3               | 37.0               | 3.59                  | 62.4                 |
>
> It should be mentioned that the average METB scores [https://huggingface.co/spaces/mteb/leaderboard] over 56 datasets for text-embedding-ada-002, bge-base-en and GritLM-7B are 60.99, 63.36, 66.76, respectively, where a better METB score means a better retriever.
>
> We observe that better performance is obtained when using a better retriever. For example, on the reaction and summarization tasks, which require retrieving detailed contexts from the memories, using better retrievers leads to more useful contexts and better performance.
>
> We will add this discussion in our new version.
>
> **Q3. Some personality or response characteristics are not well revealed.**
>
> **A3**. Thanks for your advice.
>
> It should be mentioned that the personality and characteristics of roles have been implicitly extracted in our hierarchical memory system. As shown in Fig.4(b), we provide the visualization by playing with RoleAgent "Sheldon" in the interaction stage and we observe that the RoleAgent "Sheldon" produces high-quality and interesting responses, which can reflect the personality of "Sheldon" well.
>
> Besides, following RoleLLM[R1], we propose to evaluate the response characteristics on the English subset of RoleAgentBench. Specifically, we first prepare the role profiles composed of GPT-4-generated role descriptions and catchphrases, and structured dialogues parsed from the script. Then, we use the RoleGPT approach in RoleLLM based on dialogue engineering and GPT-4 to produce the stylized answers as the stylized ground truths based on the reference answers for these questions. Then, in the following table, we evaluate the Rouge-L between model predictions and stylized ground truths, and use GPT-4 winrate to measure the response personality.
>
> Moreover, the RoleLLM implements an explicit stylized prompting strategy based on role profiles and  dialogues to enhance the response characteristics, which is also orthogonal to our RoleAgent. Therefore, in the following table, we compare with RoleLLM baseline (RoleLLM (stylized prompting)) and combine our RoleAgent with the explicit stylized prompting strategy in RoleLLM to produce the results of RoleAgent (GPT-3.5+stylized prompting in RoleLLM).
>
> In the following table, we observe that our RoleAgent (GPT-3.5) is better than RoleLLM (GPT-3.5, stylized prompting) a lot, which shows the effectiveness of improving response characteristics based on our hierarchical memory system in RoleAgent. Besides, when combined with RoleLLM, we can achieve better results. Note that we also report the results of using LLaMA3-8B and we have similar observations.
>
> |                                                     | Rouge-L | GPT-4 Win rate |
> | --------------------------------------------------- | ------- | -------------- |
> | RoleLLM (GPT-3.5, stylized prompting)              | 37.12   | 31.25          |
> | RoleAgent (GPT-3.5)                                 | 45.19   | 43.50          |
> | RoleAgent (GPT-3.5+stylized prompting in RoleLLM)   | 49.68   | 45.06          |
> | RoleLLM (LLaMA3-8B, stylized prompting)            | 26.41   | 23.83          |
> | RoleAgent (LLaMA3-8B)                               | 38.06   | 34.25          |
> | RoleAgent (LLaMA3-8B+stylized prompting in RoleLLM) | 41.48   | 36.90          |
>
> **Q4. Running code available online.**
>
> **A4**. Currently, we are organizing the code and preparing the new version of our RoleAgent based on these constructive comments from reviewers. We will release the code when our paper is released on the public platform (e.g., Arxiv).

---

### Author Rebuttal · Authors · 2024-08-17

## **General Response**

Thanks a lot for handling/reviewing our submitted manuscript. We would like to thank the reviewers for their thoughtful and constructive comments and suggestions. By addressing each of the issues raised by the reviewers, we believe that the quality and clarity of our RoleAgent can be improved a lot.

**G.Q1. General response on evaluation of the hierarchical memory system.**

**A1**. For the evaluation of memory, we refer to the survey on the memory of LLM-based agents [R1] to provide the subjective evaluation and objective evaluation on the English subset of RoleAgentBench. For subjective evaluation, we test coherence and rationality, where the human evaluation and GPT-4 evaluation are used to evaluate the top-5 memories for each question. For objective evaluation, we test the reference accuracy, where the recall is reported. The details are as follows:

**Subjective evaluation**

(1). Coherence refers to whether the recalled memory is natural and suitable for the current context, and we need to assess whether the response is naturally and coherently structured, connecting the dialogue context and retrieved memory (labels: 0: not coherent, 0.5: partially coherent, 1: coherent).

(2). Rationality evaluates whether the recalled memory is reasonable, and we need to check whether the retrieved memory contains non-factual content according to the context. (labels: 0: not rational, 0.5: partially rational, 1: rational).

The following table reports the results (%) of coherence and rationality, where we divide the summation scores on the memories by the number of memories.

|             | Coherence (Human) | Coherence (GPT-4) | Rationality (Human) | Rationality (GPT-4) |
| ----------- | ----------------- | ----------------- | ------------------- | ------------------- |
| Qwen-Max    | 82.8              | 85.9              | 84.9                | 85.2                |
| GPT-3.5     | 75.9              | 72.6              | 83.4                | 81.9                |
| Qwen1.5-14B | 66.1              | 63.2              | 78.1                | 81.4                |
| LLaMA3-8B   | 70.40             | 68.6              | 81.2                | 78.5                |

**Objective Evaluation**

We provide the reference accuracy to evaluate whether the agent can discover relevant memory contents to answer the questions. Specifically, we compare the retrieved memory with the pre-prepared ground truth to evaluate the recall. Note that we annotated the golden retrieval segments of the question based on GPT-4 and human verification.

|             | **Recall** |
| ----------- | ---------- |
| GPT-3.5     | 86.7       |
| Qwen-Max    | 87.8       |
| Qwen1.5-14B | 77.4       |
| LLaMA3-8B   | 75.3       |

[R1] A survey on the memory mechanism of large language model based agents

The prompts for GPT-4 evaluation are as follows.

### LLM-Judge Coherence Prompt

```Plain
You are a good annotator for evaluating the coherence of the memory retrieved by an Agent system. The Agent system will respond to the user's question by retrieving the relevant memory. Below are the details of the task and specific requirements.

Task Definition:
Evaluate the coherence of the memory retrieved by the Agent system. You will be provided with the user's question, the response provided by the Agent, and the retrieved memory.

Coherence Definition: Coherence aims to assess whether the retrieved memory is natural and suitable for the current context (user's question and Agent's response).

Evaluation Scale:
0 (Not Coherent): The retrieved memory is completely unrelated to the user's question and the Agent's response. It lacks logical connection and disrupts the flow of the conversation.
0.5 (Partially Coherent): The retrieved memory is somewhat related to the user's question and the Agent's response but does not fully align. It has a weak logical connection, which may cause some confusion or partial misunderstandings.
1 (Coherent): The retrieved memory is highly relevant and naturally fits the user's question and the Agent's response. It maintains a clear and logical connection, enhancing the overall flow of the conversation.

Your task is to evaluate the provided instances and output the results in JSON format.

Input Format:
{
  "Question": {question},
  "Response": {response},
  "Retrieved_Memory": {memory}
}
Output Format:
{"Coherence": {score}}
```

### LLM-Judge Rationality Score

```Plain
You are a good annotator for evaluating the rationality of the memory retrieved by an Agent system. The Agent system will respond to the user's question by retrieving the relevant memory. Below are the details of the task and specific requirements.

Task Definition:
Evaluate the rationality of the memory retrieved by the Agent system. You will be provided with the user's question and the retrieved memory.


Rationality Definition: Rationality evaluates whether the retrieved memory is reasonable and assesses whether the retrieved memory contains non-factual content according to the context.

Evaluation Scale:
0 (Not Rational): The retrieved memory is not reasonable or factually incorrect. The response includes information that is clearly wrong or illogical.
0.5 (Partially Rational): The retrieved memory is somewhat reasonable but may contain minor inaccuracies or logical flaws. The retrieved memory is not entirely incorrect but may be somewhat misleading or incomplete.
1 (Rational): The retrieved memory is entirely reasonable and factually correct. The retrieved memory is accurate and logically sound, providing appropriate and correct information.

Your task is to evaluate the provided instances and output the results in JSON format.

Input Format:
{
  "Question": {question},
  "Retrieved_Memory": {memory}
}

Output Format:
{"Rationality": {score}}
```

Note that the human evaluation guideline is similar to the GPT-4 evaluation prompt.

We will add the above evaluation on the memory system in our new version.

---

### Decision · Program_Chairs · 2024-09-26

**Decision:**

Accept (Poster)

**Comment:**

This paper propose a RoleAgent framework and collects the role agent profiles and memories from scripts. The paper introduces a comprehensive role agent benchmark covering 20 English and 5 Chinese scripts. Reviewers largely agree that the paper should be accepted and having looked through the paper it seems as though the dataset does produce different enough results across models that it would have potential as an LLM benchmarking tool.